# Development of Marker-Based Motion Capture Using RGB Cameras: A Neural Network Approach for Spherical Marker Detection

**DOI:** 10.3390/s25175228

**Published:** 2025-08-22

**Authors:** Yuji Ohshima

**Affiliations:** Faculty of Human Health, Kurume University, 1635 Miichou, Kurume 839-0851, Fukuoka, Japan; ohshima_yuuji@kurume-u.ac.jp

**Keywords:** human motion measurement, YOLO, object detection

## Abstract

Marker-based motion capture systems using infrared cameras (IR MoCaps) are commonly employed in biomechanical research. However, their high costs pose challenges for many institutions seeking to implement such systems. This study aims to develop a neural network (NN) model to estimate the digitized coordinates of spherical markers and to establish a lower-cost marker-based motion capture system using RGB cameras. Thirteen participants were instructed to walk at self-selected speeds while their movements were recorded with eight RGB cameras. Each participant undertook trials with 24 mm spherical markers attached to 25 body landmarks (marker trials), as well as trials without markers (non-marker trials). To generate training data, virtual markers mimicking spherical markers were randomly inserted into images from the non-marker trials. These images were then used to fine-tune a pre-trained model, resulting in an NN model capable of detecting spherical markers. The digitized coordinates inferred by the NN model were employed to reconstruct the three-dimensional coordinates of the spherical markers, which were subsequently compared with the gold standard. The mean resultant error was determined to be 2.2 mm. These results suggest that the proposed method enables fully automatic marker reconstruction comparable to that of IR MoCap, highlighting its potential for application in motion analysis.

## 1. Introduction

Motion capture using cameras, a fundamental and frequently employed measurement technique in biomechanical research, relies on image analysis methods. The history of these methods dates back to 1878 when photographer Eadward Muybridge captured the motion of horses, revealing the existence of a flight phase in a horse’s gallop through sequential photographs [1]. In 1882, physiologist Etienne-Jules Marey developed a camera known as the “chronophotographic rifle,” capable of taking 12 images per second. Marey used this innovation to capture various motions, including bird wing flapping, cat landing maneuvers (the cat righting reflex), and human jumping movements [2]. By around 1930, two-dimensional motion analysis of body movements began to develop [3,4,5]. For instance, Fenn [5] estimated the energy generated by muscles during running based on changes in the mechanical energy of body segments. The introduction of the Direct Linear Transformation (DLT) method by Abdel-Aziz and Karara [6] marked a significant advancement, leading to the widespread adoption of three-dimensional motion analysis [7,8,9,10,11,12]. The DLT method continues to be utilized in recent studies [13], reconstructing three-dimensional coordinates by capturing measurement points with multiple cameras and obtaining coordinate values from each camera’s image plane (digitized coordinates). However, when employing the DLT method for motion analysis, RGB cameras are often used, necessitating the manual acquisition of digitized coordinates for body landmarks. This digitization process is time-consuming and labor-intensive, posing a significant challenge in accumulating three-dimensional motion data.

Infrared-camera-based motion capture systems (IR MoCap) have emerged as a solution to the challenges associated with the digitizing process. IR MoCap can automatically acquire the digitized coordinates of reflective markers attached to the body, enabling the reconstruction of three-dimensional marker coordinates without manual digitization. Compared to motion capture systems that use inertial measurement units, IR MoCap collects three-dimensional motion data with significantly higher accuracy [14] and is currently the most widely used motion capture method. However, IR MoCap has limitations; it cannot be employed in situations where reflective markers cannot be attached to the body, such as during movements involving large contact areas between the body and the ground, or in competitive environments. Additionally, implementing IR MoCap necessitates expensive infrared cameras and specialized software, which can be prohibitive for many research and educational institutions [15].

As described previously, motion analysis using RGB cameras presents challenges related to the digitization process, while IR MoCap poses issues concerning financial costs. Recently, many low-cost, high-performance RGB cameras have become commercially available. If the digitized coordinates of the markers attached to the body can be automatically extracted from images captured by RGB cameras, both issues can be resolved. To reduce the labor associated with the digitization process, Figueroa et al. [16] developed a method for automatically acquiring marker coordinates by utilizing the pixel information around the marker location in the previous frame. This approach alleviates the burden of manual digitization. However, for frames where no previous frame exists, such as the first frame of a video, manual digitization is still required. Thus, the method developed by Figueroa et al. [16] does not fully automate digitization. Currently, no method has been developed that enables the fully automatic acquisition of digitized coordinates for spherical markers from RGB images in the same manner as IR MoCap. On the other hand, recent developments in motion capture technology have introduced systems that use neural network (NN) models to automatically infer digitized coordinates of anatomical landmarks from RGB images—referred to as markerless motion capture (Markerless MoCap) systems. Studies have validated the accuracy of such systems [13,17,18]. For example, Kanko et al. [18] evaluated the accuracy of Theia 3D (Theia Markerless Inc., Kingston, ON, Canada), a commercially available Markerless MoCap system designed for biomechanical analysis, and reported that the positional error of joint centers was approximately 20–30 mm compared to IR MoCap, indicating sufficient accuracy for motion analysis. However, the same study also noted that the accuracy of joint rotation angles around the segment’s longitudinal axis—such as hip internal/external rotation—was relatively low. Moreover, the high cost associated with systems like Theia 3D remains a significant limitation. Thus, while motion capture using RGB cameras has made considerable progress, there remain unresolved challenges in terms of measurement accuracy and system cost. To address these limitations, the development of a marker-based motion capture system using RGB cameras (RGB MoCap) is considered meaningful for the accumulation of high-quality biomechanical data in a more accessible and cost-effective manner.

In recent years, various NN models have been developed for different applications. Among these, NN models infer the locations and sizes of objects within images (object detection; see Figure 1). Several types of object detection NN models have been developed [19,20,21]. For example, YOLOv9, created by Wang et al. [20], is publicly available on GitHub (https://github.com/WongKinYiu/yolov9), allowing anyone worldwide to use it freely. Furthermore, YOLOv9 can detect objects not recognized by the pre-trained model by updating the NN model weights through fine-tuning, and a program code for fine-tuning is also provided. Therefore, if marker detection on the body can be achieved through fine-tuning, the digitization process can be fully automated, enabling the development of RGB MoCap.

The purpose of this work is to develop an NN model that estimates the digitized coordinates of spherical marker centers and to evaluate the accuracy of the reconstructed three-dimensional marker coordinates based on the estimated digitized coordinates. If the proposed method achieves high accuracy in reconstructing the three-dimensional marker coordinates, motion capture similar to IR MoCap would become feasible using RGB cameras, thereby simplifying the collection of three-dimensional motion data.

The main contributions of this study are summarized as follows:Development of a neural network model to infer digitized coordinates of spherical markers from RGB images.Validation of the 3D reconstruction accuracy of markers based on the inferred digitized coordinates.Demonstration that the proposed method enables infrared motion capture (IR MoCap) level accuracy using only RGB cameras, simplifying 3D motion data acquisition.

The remainder of this paper is organized as follows:Section 2 describes the experimental procedures, the fine-tuning process, and the accuracy evaluation.Section 3 presents the results of the accuracy evaluation and robustness evaluation.Section 4 mainly discusses the accuracy of the proposed method and provides a comparison with IR MoCap and Markerless MoCap.Section 5 presents the conclusions of this study.

## 2. Methods

### 2.1. Data Collection

The participants included 13 males (height: 1.71 ± 0.06 m, weight: 69.3 ± 9.3 kg, age: 20.6 ± 0.8 years). This study received approval from the Research Ethics Committee of the Mii Campus at Kurume University. Before the experiments, each participant was given an explanation of the study’s objectives and tasks, and informed consent was obtained both verbally and in writing.

Each participant was instructed to walk 15 m twice at a self-selected speed. Motion from 8 to 12 m after the starting position was recorded using 8-RGB cameras (Lumix BGH1, Panasonic Corporation, Osaka, Japan; frame rate: 240 fps, shutter speed: 1/500 s), which were synchronized using a time synchronization unit (Ultra Sync ONE, Timecode Systems Ltd., Worcestershire, UK), as illustrated in Figure 2. Prior to the experimental trials, 24 mm spherical markers (Figure 3) were attached to 25 anatomical landmarks based on the definitions provided by Suzuki et al. [22], as shown in Figure 4. Due to the limited number of available cameras, it was not possible to capture all anatomical landmarks defined by Suzuki et al. [22]. Therefore, markers were only attached to landmarks on the left side of the body for accuracy verification. After completing the two walking trials, all spherical markers were removed, and participants were instructed to perform one additional walking trial. This trial was recorded to create training data for fine-tuning. In this study, the trial with markers attached is referred to as the “marker trial,” while the trial without markers is referred to as the “non-marker trial.” The analysis concentrated on the gait cycle from left foot contact to the subsequent left foot contact.

### 2.2. Development of Neural Network Model for Detecting Spherical Markers

In this study, a model for detecting spherical markers was developed by fine-tuning the YOLOv9 neural network model [20]. The training data were generated by randomly inserting images of virtual markers that mimicked spherical markers into images from non-marker trials, as shown in Figure 5. Since the spherical markers were fixed to the body using elastic adhesive tape during the experiments (Figure 3), virtual images of the tape and spheres were generated and combined to represent the markers attached to the body (virtual markers). Additionally, nearest-neighbor interpolation was used to resize the images of the virtual tape and spheres.

#### 2.2.1. Generation of Virtual Spheres

The spherical markers used in the experiment were not perfect spheres and thus appeared as distorted circles in the captured images. To account for this, various shapes approximating circles were generated. First, as shown in Figure 6a, a circular image was divided into nine equal regions using four lines (Line 1 to Line 4) as boundaries. The positions of these lines were randomly determined within a predefined range (Figure 6a), and the sizes of the respective image regions were modified accordingly to produce deformed shapes that approximated circles (Figure 6b). Furthermore, considering the possibility that the actual spherical markers may resemble ellipsoids, the vertical or horizontal dimensions of each image in Figure 6b were randomly scaled within a range of 80–100%, resulting in additional variations in shape (see Figure 6c). These images were used as virtual spherical markers to fine-tune the model. In addition, the color of the virtual markers was randomized in terms of hue (H: 140–160), saturation (S: 240–255), and lightness (L: 140–170), all on a 0–255 scale, within ranges empirically determined as follows: The markers used in the experiments were recorded using a video camera, and the resulting footage was transferred to a personal computer. The HSL values of the markers were extracted from the images, and the ranges for each parameter were subjectively determined based on these observations.

#### 2.2.2. Generation of Virtual Tape

Figure 7a shows a schematic of the virtual tape. The length, width, position, rotation angle, and whether the tape was rendered (with a 50% probability) were randomly determined to generate virtual tape images, as illustrated in Figure 7b. To simulate the scenario in which the camera’s optical axis was not perpendicular to the tape surface, the vertical size of the image (defined as the aspect ratio of the virtual tape) was randomly varied within a range of 10% to 100%, resulting in images such as those shown in Figure 7c. Because the adhesive tape used in this study was white, the color of the virtual tape was randomly assigned within specified ranges of hue (0–30), saturation (0–20), and lightness (200–255), all on a 0–255 scale.

#### 2.2.3. Combination of Virtual Sphere and Virtual Tape Images

By combining the randomly generated virtual sphere and virtual tape images, we generated images that mimicked spherical markers affixed to the body. However, the positional relationship between the tape and sphere on the image plane depends on the camera’s orientation relative to the spherical marker during image capture. Thus, as shown in Figure 8, the position of the virtual tape image relative to the virtual sphere image was determined using Equation (1), as follows:(1)$$v_{1}=v_{2}\cdot r_{aspect}+v_{3}\cdot \left(1-r_{aspect}\right)$$
where $v_{2}$ represents the digitized coordinates of the centroid of the virtual sphere, $v_{3}$ represents the digitized coordinates of the bottom edge of the sphere, and $r_{aspect}$ indicates the aspect ratio of the virtual tape. The two images were then combined such that the center of the virtual tape image coincided with $v_{1}$. Subsequently, the image size of the virtual marker was randomly adjusted so that the diameter of the virtual sphere ranged from 8 to 24 pixels. The resulting image was then randomly rotated and inserted into a non-marker trial image.

#### 2.2.4. Insertion of Virtual Markers into Non-Marker Trial Images and Blurring Process

In YOLOv9, the standard input image size for fine-tuning is 640 × 640 pixels. As shown in Figure 9a–c, a 320 × 320 pixel region was randomly cropped from the video frame of a non-marker trial and then resized to 640 × 640 pixels using bicubic interpolation (Figure 9d). To generate the training images, 7 to 12 virtual marker images were randomly inserted into the resized images (Figure 9e). Since the bounding boxes for all virtual spherical markers were predefined, annotation files were automatically generated. However, the boundaries of the inserted virtual markers often appeared visually unnatural. To address this, a Gaussian filter was applied to smooth the pixel transitions at the marker edges. The filter had a mean of 0, and the standard deviation was randomly determined for each marker within the range of 0.5 to 3.0. After filtering, the pixel values outside the marker regions were retained from the original non-marker trial image.

#### 2.2.5. Fine-Tuning

Fine-tuning was performed using a pre-trained YOLOv9 model (yolov9-t-converted.pt) and a training script (train.py). Training was conducted for 2000 epochs, with 18 training images generated per epoch using the method described in the previous section (batch size = 18). The hyperparameters (hyp.scratch-high.yaml), data augmentation and layer freezing were not applied, while the other parameters were maintained at their default values. Fine-tuning was performed separately for each subject and camera on an NVIDIA RTX 3090 GPU (NVIDIA Corporation, Santa Clara, CA, USA), with the training time per run being approximately 4 min. Since the proposed method is intended for use in controlled experimental environments where researchers collect their own data, this subject- and camera-specific fine-tuning process does not pose a practical issue.

#### 2.2.6. Detection of Spherical Markers

Using the model obtained through fine-tuning and the detection script (detect.py), spherical markers within the marker trial images were detected, and their digitized coordinates were obtained. The user-defined detection parameters were set as follows: both the confidence threshold and Intersection over Union (IoU) threshold were set to 0.01. Although the resolution of the recorded video was 1080 × 1920 pixels, the images were upscaled to 2160 × 3840 pixels using bicubic interpolation prior to marker detection. The inference time per image was approximately 20 ms on an NVIDIA RTX 3090 GPU.

### 2.3. Verification of Accuracy of 3D Coordinates

#### 2.3.1. Estimation of Camera Parameters

Multiple cameras were used to synchronously capture the motion of the calibration wand (Figure 10), which moved randomly within the measurement area. The camera parameters were estimated based on the known distance between the spherical markers attached to the wand and their digitized coordinates [23,24]. These digitized coordinates were obtained using an NN model trained for spherical marker detection. The fine-tuning process followed the same procedure as the marker trials. However, a new NN Model was created to detect both types of markers, as the wand was equipped with both blue and green spherical markers. The hue, saturation, and lightness ranges for the virtual spheres were set separately for the two colors: for the green markers, the hue was set to 100–120, saturation to 200–255, and lightness to 30–100; for the blue markers, the hue was set to 145–165, saturation to 200–255, and lightness to 30–100. Virtual tape was not used. For marker detection, the confidence threshold and IoU threshold were set to 0.50 and 0.01, respectively. Because the spherical markers on the wand were larger than those attached to the body, the original resolution of the recorded images (1080 × 1920 pixels) was used as the input. The 3D coordinates of the spherical markers on the wand were reconstructed using the estimated camera parameters. The average error in the distance between the two markers was 0.1 mm. Furthermore, the shortest average distance between the camera ray (defined as a half-line computed from the digitized coordinates and camera parameters) and the reconstructed 3D coordinates (referred to as the reconstruction error) was 1.8 mm.

#### 2.3.2. Reconstruction of 3D Marker Positions Based on Proposed Method

By estimating the camera parameters, it is possible to reconstruct the 3D coordinates of spherical markers from their digitized coordinates obtained from multiple cameras. However, the correspondence between the markers detected by each camera was not known in advance. To address this issue, 3D coordinates were reconstructed using data from at least three cameras. For each candidate point, combinations were explored to ensure that the distance between the reconstructed point and the camera ray was less than 5 mm for all cameras. Marker correspondences across the cameras were then established, allowing for the reconstruction of the 3D positions of the spherical markers. Since the correspondence between the reconstructed points and the physical markers attached to the subject was also unknown, a graphical user interface (GUI) was developed using numerical analysis software (MATLAB 2021b; MathWorks Inc., Natick, MA, USA) to manually label the markers. A total of 41,275 points (analysis points × frames) were expected to be reconstructed; however, 100 points did not meet the condition of being reconstructed using three or more cameras.

#### 2.3.3. Determination of Gold Standard

To determine the gold standard of 3D coordinates, the digitized coordinates of the centers of the spherical markers attached to the subject were manually obtained using a custom-built digitizer developed with numerical analysis software (MATLAB 2021b; MathWorks Inc., Natick, MA, USA). The 3D coordinates were then reconstructed from the estimated camera parameters and digitized coordinates, and these reconstructed coordinates were defined as the gold standard. The reconstruction utilized an average of 6.7 cameras, and all 41,275 points were successfully reconstructed, yielding a mean reconstruction error of 1.52 mm.

### 2.4. Accuracy Evaluation of Proposed Method

The error was defined as each component of the vector derived from the marker coordinates reconstructed by the proposed method compared to the gold standard. Accuracy was evaluated using Bland–Altman analysis. The presence of fixed bias was examined through a paired *t*-test to determine whether the mean error was zero. Proportional bias was assessed using simple linear regression analysis, with the error as the dependent variable and the mean of the two measurements as the independent variable, testing whether the slope of the regression line was zero with a *t*-test. Parametric analyses were employed in this study due to the large sample size (41,175 samples) and the unimodal distribution of the error histogram (see Figure 11). The length of the vector is defined as the resultant error.

### 2.5. Robustness Evaluation of the Proposed Method

To evaluate the robustness of the proposed method, the following procedure was performed:A spherical marker was randomly selected.For the selected marker, random noise was added to the digitized coordinates detected by the NN model, and the 3D coordinates were reconstructed using these noisy coordinates. The magnitude of the noise was set such that it falls within a circular area obtained by reprojecting a sphere with a diameter of 12 mm (half the size of the spherical marker) centered at the 3D coordinates reconstructed by the proposed method onto the image plane.The distance between the reconstructed 3D coordinates obtained in step 2 and those reconstructed by the proposed method was calculated.Steps 1 to 3 were repeated 1,000,000 times to obtain the average and maximum distances.

Since the number of cameras used for reconstruction varies depending on the spherical marker (433 markers with 3 cameras, 4743 with 4 cameras, 7205 with 5 cameras, 6828 with 6 cameras, 8159 with 7 cameras, and 13,807 with 8 cameras), the above procedure was performed for each camera count.

## 3. Results

Table 1 and Figure 12 present the results of the Bland–Altman analysis. Significant fixed biases were observed in all components, with mean values of −0.192 mm, −0.402 mm, and −0.467 mm for the x-, y-, and z-components, respectively. Additionally, significant proportional biases were identified in the x- and z-components, with regression slopes of 0.000176 and −0.000437, respectively. Figure 13 displays a histogram of the resultant error, with a mean value of 2.24 mm, a 95th percentile of 4.58 mm, and a 99th percentile of 7.07 mm.

Table 2 presents the results of the robustness evaluation. When three cameras were used for reconstruction, the average distance was 5.23 mm. Furthermore, the average distance decreased as the number of cameras increased, reaching 2.43 mm when eight cameras were used.

## 4. Discussion

The aim of this study was to develop a neural network (NN) model for detecting spherical markers, reconstructing 3D coordinates based on the inferred digitized positions of the marker centers, and evaluating the accuracy of the reconstruction. The gold standard in this study was established using an average of 6.7 cameras, with a mean reconstruction error of 1.52 mm. In infrared motion capture (IR MoCap), where markers are attached to the body surface, measurement accuracy is often compromised due to soft tissue artifacts, with average errors reported to range from 5 to 15 mm [25]. Furthermore, even when anatomical landmarks on the body surface can be precisely identified, it is challenging for an examiner to attach the marker to the exact same location during each measurement. Considering these factors, the gold standard used in this study can be regarded as highly reliable for validating the accuracy of the RGB MoCap developed for motion analysis.

The Bland–Altman analysis of the reconstructed 3D coordinates of the spherical markers revealed fixed biases; however, their mean values were −0.192 mm, −0.402 mm, and −0.467 mm for the x-, y-, and z-components, respectively, all remaining below 1 mm (see Table 1). Proportional biases were also observed in the x- and z-components, but the regression slopes were relatively small, at 0.000176 and −0.000437, respectively (see Table 1). The impact of these proportional biases was estimated to be less than 1 mm, considering the measurement volume of 2 m × 4 m × 2 m. These results suggest that both fixed and proportional biases are unlikely to significantly influence motion analysis outcomes.

The mean resultant error of the RGB MoCap system was 2.24 mm (Figure 13). In comparison, markerless MoCap systems have been reported to show joint center position errors of approximately 20–30 mm during gait analysis [13,17,18]. Additionally, the accuracy of marker positions in IR MoCap systems has been reported to be less than 1 mm [26]. Based on the current results and related literature, while markerless MoCap offers the advantage of allowing natural movement without the need for marker attachment, it may not be suitable in situations where high-precision measurements are required. In cases where the use of IR MoCap is not feasible, the RGB MoCap developed in this study can be considered a useful alternative in terms of both accuracy and practicality. Furthermore, although RGB MoCap is less accurate than IR MoCap, it has the advantage of being easily integrated with markerless motion capture systems. Since markerless MoCap cannot track objects such as rackets or bats, simultaneous analysis of body and equipment movement has traditionally required the combined use of IR MoCap. By using the RGB MoCap developed in this study as a substitute for IR MoCap, integrated motion analysis of both the body and equipment can be performed in combination with markerless MoCap, without relying on infrared cameras. This approach offers significant benefits in terms of simplifying measurement equipment and reducing costs.

Based on the results of the robustness evaluation, it was shown that the reconstruction of 3D coordinates becomes more robust as the number of cameras increases (see Table 2). Since the digitized coordinates obtained always contain errors of unknown magnitude, it is desirable to prepare as many cameras as possible and arrange them appropriately during the experiment. Furthermore, as shown in Figure 14, in the proposed NN model, when a part of the marker is occluded, digitized coordinates shifted from the marker’s center may be detected, which can lead to a decrease in the accuracy of the 3D coordinates. Therefore, by excluding markers whose bounding box shapes significantly deviate from a square from reconstruction, it may be possible to suppress a decrease in accuracy.

As a future challenge, this study used relatively large spherical markers (diameter: 24 mm). Thus, it is necessary to investigate whether smaller markers can also be reliably detected, and to determine the minimum marker size required for successful detection. Additionally, it is important to verify whether the accuracy can be maintained not only for walking motions but also for other bodily movements. As a limitation of the proposed method, real-time measurement is likely to be challenging because fine-tuning must be performed for each camera and subject.

## 5. Conclusions

This study proposed a method for constructing a neural network model to detect spherical markers in images and developed an RGB camera-based motion capture system (RGB MoCap) that reconstructs the 3D coordinates of markers from inferred digitized coordinates. The results indicated that the mean resultant error of the 3D coordinates was 2.24 mm, demonstrating sufficient accuracy for general motion analysis.

## 6. Patents

The author intends to file a patent application related to the method described in this manuscript.

## Figures and Tables

**Figure 1 sensors-25-05228-f001:**
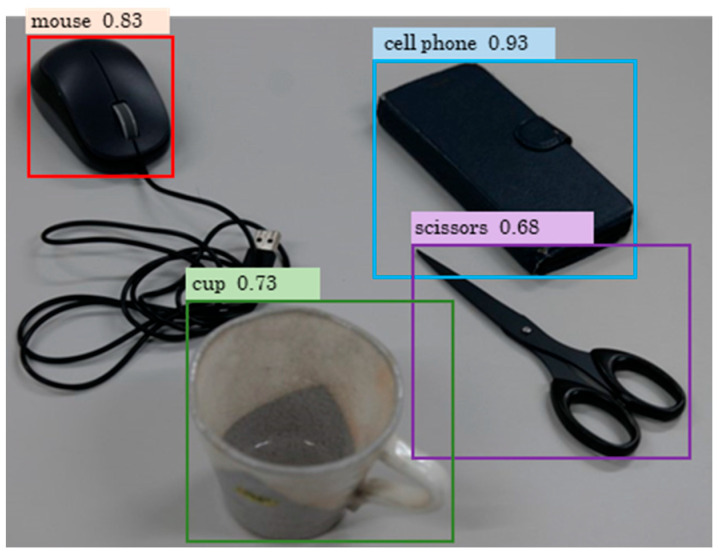
Example of object detection using YOLOv9.

**Figure 2 sensors-25-05228-f002:**
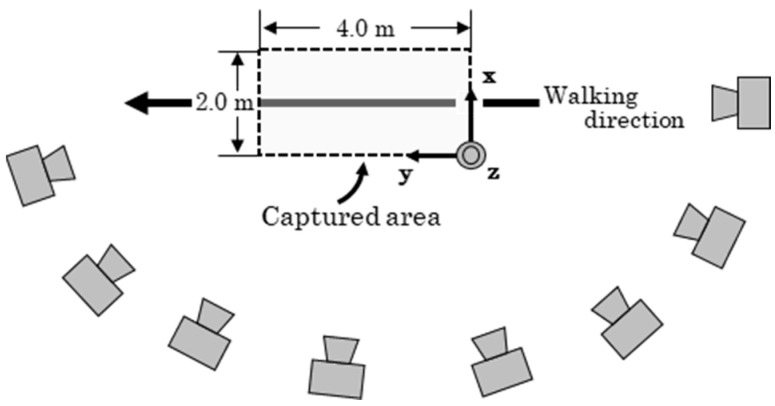
Experimental setup. Eight RGB cameras were arranged to cover a measurement volume of 2.0 m (x) × 4.0 m (y) × 2.0 m (z).

**Figure 3 sensors-25-05228-f003:**
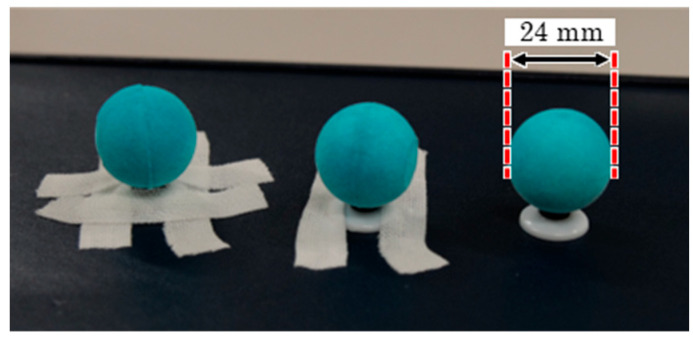
Spherical markers used in the experiment.

**Figure 4 sensors-25-05228-f004:**
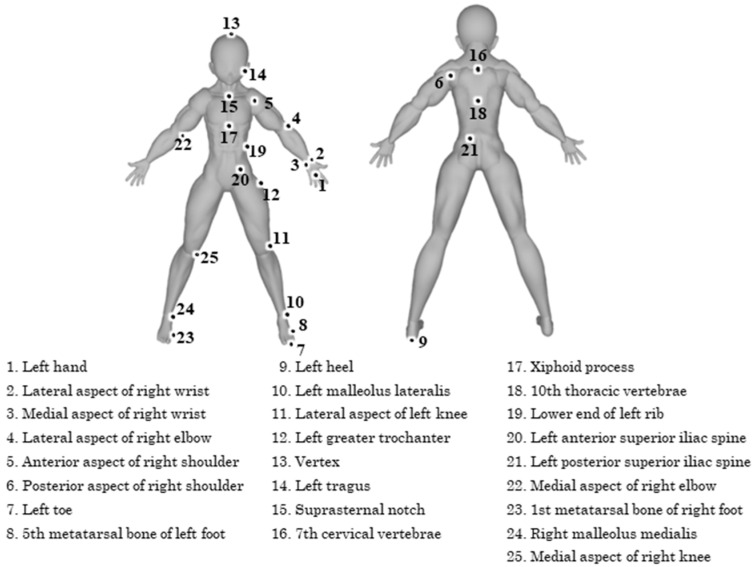
Landmarks with markers affixed.

**Figure 5 sensors-25-05228-f005:**
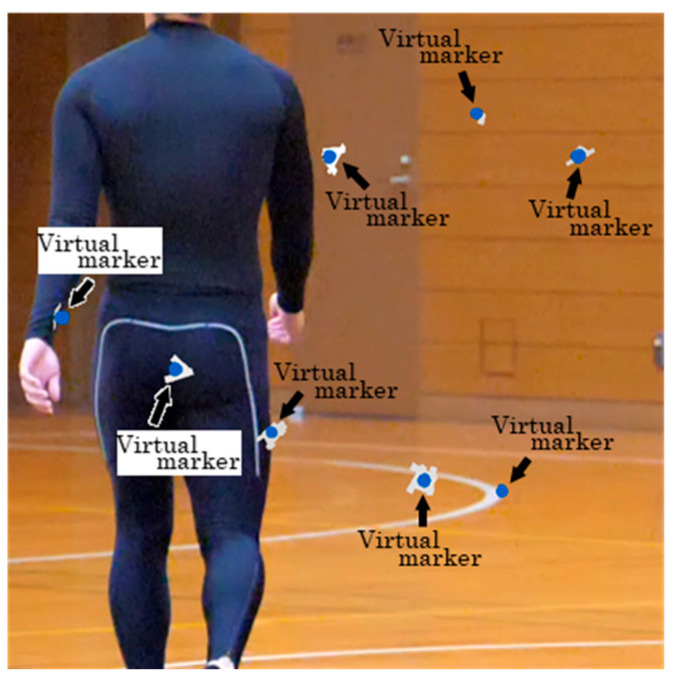
Example of image used for fine-tuning.

**Figure 6 sensors-25-05228-f006:**
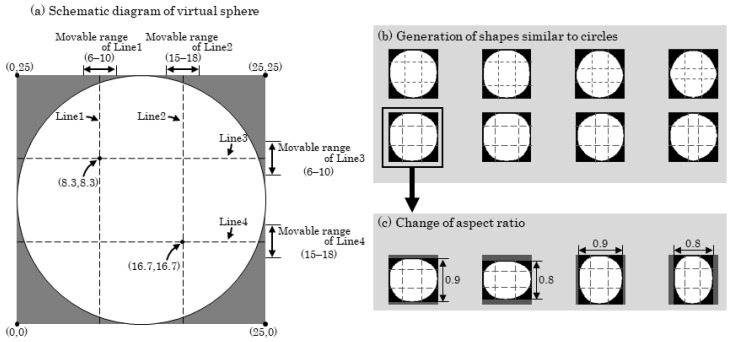
Procedure for virtual sphere generation. (**a**) Schematic diagram of the virtual sphere; (**b**) Generation of shapes similar to circles; (**c**) Change of aspect ratio.

**Figure 7 sensors-25-05228-f007:**
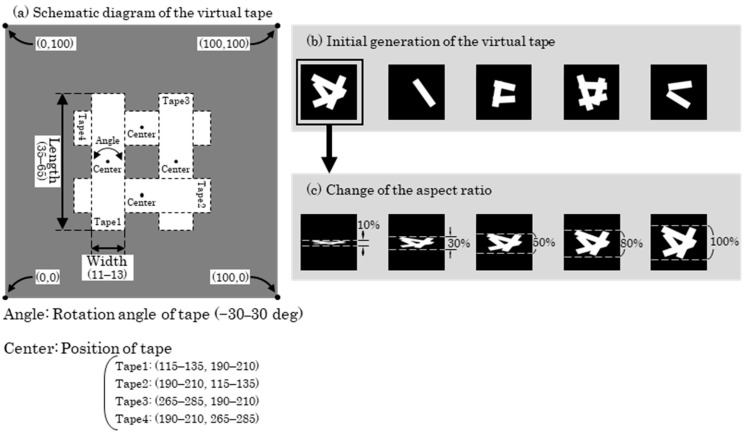
Procedure for virtual tape generation. (**a**) Schematic diagram of the virtual tape; (**b**) Initial generation of the virtual tape; (**c**) Change of the aspect ratio.

**Figure 8 sensors-25-05228-f008:**
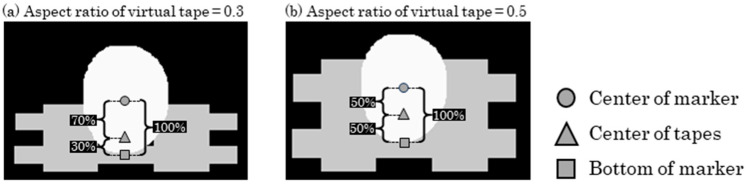
Examples of images combining a virtual sphere and virtual tapes. The gray area indicates the region of the virtual tape, and the white area indicates the region of the virtual sphere. (**a**) Virtual markers with an aspect ratio of 0.3. (**b**) Virtual markers with an aspect ratio of 0.5.

**Figure 9 sensors-25-05228-f009:**
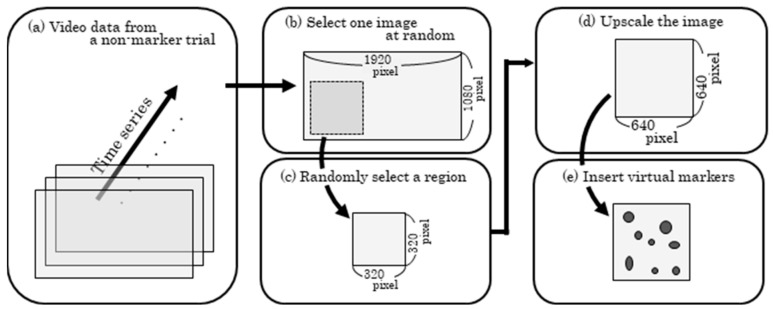
Procedure for inserting virtual markers into training images. (**a**) Video data obtained from a non-marker trial; (**b**) One frame randomly selected from the video; (**c**) A region randomly selected from the image. (**d**) The selected region is upscaled. (**e**) Virtual markers are placed on the upscaled region.

**Figure 10 sensors-25-05228-f010:**
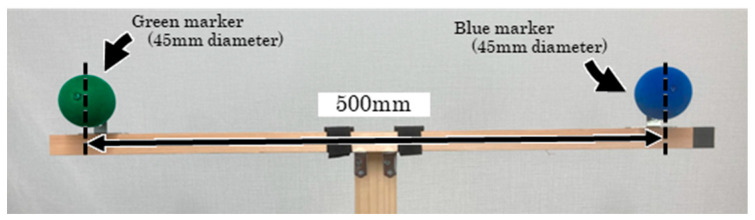
Calibration wand.

**Figure 11 sensors-25-05228-f011:**
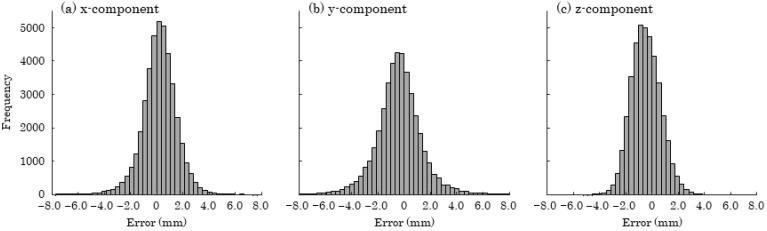
Histogram of coordinate errors. (**a**) x-component; (**b**) y-component; (**c**) z-component.

**Figure 12 sensors-25-05228-f012:**
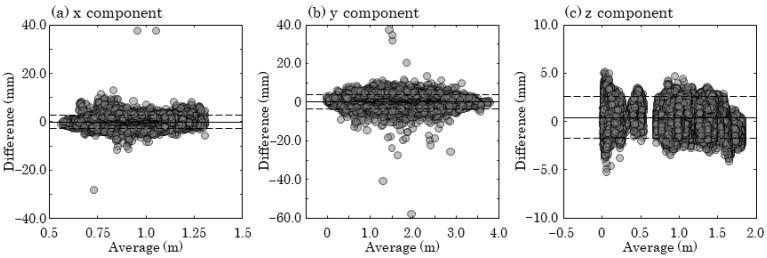
Bland–Altman plot showing the average and difference in 3D coordinates between the proposed method and gold standard. Limits of agreement are indicated as average difference (solid line) ± 1.96 times the standard deviation of difference (dashed lines). (**a**) x-component; (**b**) y-component; (**c**) z-component.

**Figure 13 sensors-25-05228-f013:**
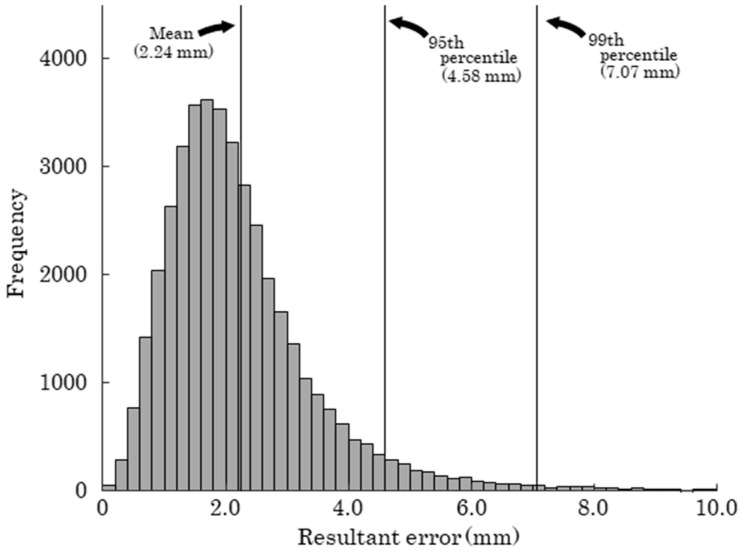
Histogram of resultant error.

**Figure 14 sensors-25-05228-f014:**
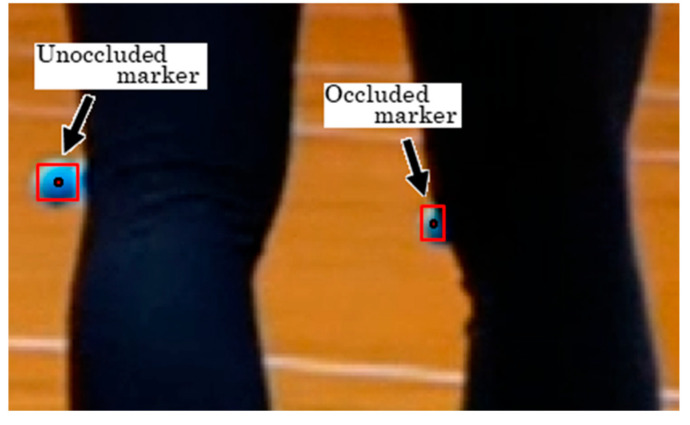
Example of marker detection with partial occlusion.

**Table 1 sensors-25-05228-t001:** Statistical evaluation of fixed and proportional bias in Bland–Altman analysis.

Component	Fixed Bias (Mean ± SD)	Proportinal Bias (Slope)
X	−0.192 ± 1.40 mm **	0.000176 *
Y	−0.402 ± 1.86 mm **	−0.000251
Z	−0.467 ± 1.11 mm **	−0.000437 **

Statistical significance: ** *p* < 0.001, *: *p* < 0.01.

**Table 2 sensors-25-05228-t002:** Robustness evaluation results.

Number of Cameras	Average Distance (mm)	Maximum Distance (mm)
3	5.23	24.68
4	4.06	15.45
5	3.42	13.53
6	2.91	10.61
7	2.61	8.77
8	2.43	7.66

## Data Availability

The data presented in this study are available upon request from the corresponding author.

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
