# Peer review of "Development of Marker-Based Motion Capture Using RGB Cameras: A Neural Network Approach for Spherical Marker Detection"

_sensors, 2025, doi:10.3390/s25175228_

Round 1
Reviewer 1 Report
Comments and Suggestions for Authors
The topic is intresting, however some points need improvement:
- The authors may consider discussing related works such as RGB-D-based vision systems for robot navigation (e.g., 'Smart robot navigation using RGB-D camera'), which highlight how depth-enhanced RGB systems are utilized in real-time detection and motion analysis.
This could broaden the comparison and application scope of the proposed RGB MoCap approach.
- The YOLOv9 model is fine-tuned using artificially generated virtual markers, which may not fully represent real-world variations in lighting, occlusions, or body movement.
- The paper does not compare its RGB MoCap approach with other state-of-the-art RGB-based motion capture methods, limiting the context of its performance claims.
- The error evaluation is limited to mean and percentile metrics; no robustness analysis is performed under challenging conditions (e.g., partial marker occlusion, camera misalignment).
- The experiments are restricted to walking tasks, leaving uncertainty about the system’s performance on more complex or dynamic motions.
- Fine-tuning is done separately for each subject and camera, which raises concerns about scalability and general applicability without subject-specific training.
- The proposed approach relies on relatively large markers (24 mm), and the paper does not provide experimental evidence of performance with smaller or less visible markers.
- Have the authors considered using explainable AI techniques, such as SHAP or LIME, to analyze the YOLOv9 model's predictions? This could provide insights into which visual features are
most critical for marker detection and improve the interpretability of the results.
- While the conclusion mentions potential future challenges, the discussion lacks a critical reflection on computational costs, training time, or real-time performance feasibility.
- The reference list relies heavily on older studies (e.g., Fenn 1930, Elftman 1940). While these works are historically relevant, the paper would benefit from citing more recent studies (2020–2025) on marker-based and markerless motion capture, deep learning for pose estimation,
and RGB/RGB-D tracking to reflect the current state of research.
Reviewer 2 Report
Comments and Suggestions for Authors
The manuscript entitled “Development of Marker-Based Motion Capture Using RGB Cameras: A Neural Network Approach for Spherical Marker Detection” presents a novel and practical approach to motion capture by employing a fine-tuned YOLOv9 model for automatic detection of spherical markers in RGB video, followed by 3D reconstruction via multi-camera triangulation. The study is well-motivated, technically sound, and addresses a critical need for low-cost motion capture alternatives to traditional IR MoCap systems. But I found some minor errors that the authors need to correct.
(1) In Line 87, the full name of NN should be provided here.
(2) In Line 103, the authors said they adopted 8-RGB cameras. However, there are just 7 cameras presented in Figure 2.
(3) In Line 146, a hyphen is redundant, "virtu-al".
(5) In Line 271, some extra words appear in the manuscript due to typographical errors.

Reviewer 3 Report
Comments and Suggestions for Authors
Title: Development of Marker-Based Motion Capture Using RGB Cameras: A Neural Network Approach for Spherical Marker Detection.
This study aims to develop a neural network (NN) model to estimate the digitized coordinates of spherical markers and to establish a marker-based motion capture system using RGB cameras. The work is interesting, and the authors have presented their work in detail. However, there are some comments worth examining carefully, as follows.
Remarks to the Authors: Please see the full comments.
- In the Abstract, it is mentioned that “However, their high costs pose challenges…..”. So, the main problem is the high cost in the existing Marker-based motion capture systems. This point should be mentioned in the defining of the proposed work to present the new idea that is added to this field.
- In the introduction, the paragraph from lines 61 to 73 are duplicated on lines 74 to 86. Please correct this problem.
- The contribution of the proposed work should be listed clearly.
- The structure of the paper should be provided for further organization.
- The introduction section provides a comprehensive foundation on the topic of this research based on various works; however, some other recent works need to be added to this section with much in-depth discussion.
- Any information, graph, equation, or dataset taken from a previous source must be documented with a reliable source, unless it belongs to the authors. Please check this issue for the entire manuscript.
- Figure 1 appears to be unreferenced in the text. Please check and cite the figure in the text.
- It is recommended to write the conclusion section in a separate paragraph.
Round 2
Reviewer 1 Report
Comments and Suggestions for Authors
The paper has been improved